# Molecular memory with downstream logic processing exemplified by switchable and self-indicating guest capture and release

Brian Daly[1], Thomas S. Moody[1], Allen J.M. Huxley[1], Chaoyi Yao[1], Benjamin Schazmann[1], Andre Alves-Areias[1], John F. Malone[1], H.Q. Nimal Gunaratne[1], Peter Nockemann[1] & A. Prasanna de Silva[1]

Molecular-logic based computation (MLBC) has grown by accumulating many examples of combinational logic gates and a few sequential variants. In spite of many inspirations being available in biology, there are virtually no examples of MLBC in chemistry where sequential and combinational operations are integrated. Here we report a simple alcohol-ketone redox interconversion which switches a macrocycle between a large or small cavity, with erect aromatic walls which create a deep hydrophobic space or with collapsed walls respectively. Small aromatic guests can be captured or released in an all or none manner upon chemical command. During capture, the fluorescence of the alcohol macrocycle is quenched via fluorescent photoinduced electron transfer switching, meaning that its occupancy state is self-indicated. This represents a chemically-driven RS Flip-Flop, one of whose outputs is fed into an INHIBIT gate. Processing of outputs from memory stores is seen in the injection of packaged neurotransmitters into synaptic clefts for onward neural signalling. Overall, capture-release phenomena from discrete supermolecules now have a Boolean basis.

[1] School of Chemistry and Chemical Engineering, Queen's University, Belfast, Northern Ireland BT9 5AG, UK. Correspondence and requests for materials should be addressed to A.P.d.S. (email: a.desilva@qub.ac.uk)

Molecular-logic-based computation[1–23] currently largely consists of combinational logic arrays, with history-dependent sequential logic systems being less common and integrations of these two being rarer still[19–21]. Indeed, a molecular memory carrying a downstream processor is unknown. Nevertheless, inspiration can be found in stored neurotransmitters being released into the synaptic cleft[22,23] for further neural processing of the original signal. Although molecular versions of computer memories or RS flip-flops[24–32] are known, all-chemical versions where the device, inputs and outputs are chemical in nature are unavailable.

Reversible atom capture/release is well-established[33–35]. Molecules can also be imprisoned/liberated[36–44] but it is harder to achieve in a reversible all-or-none manner without complications of side-reactions[40] or incomplete conversion. On the other hand, related intramolecular studies are available as parts of molecular machines[5,6,45–49].

We now present a molecular memory carrying a downstream processor. In this case, it allows us to introduce an all-chemical RS flip-flop. This permits logically controlled reversible capture/release of molecules both in aqueous solution and in the solid state. This pushes the relevance of molecular-logic-based computation[1–23] to previously unaffected areas of chemistry, e.g. cleanup of pollutants, generation of chemical signals[50] and programmed release of functional species from discrete supermolecules. Logical release (but not capture) from materials (but not discrete molecules) is available[51]. Digitally capturing/releasing a molecular guest from a molecular host due to molecular inputs shows the ability to create sequential logic devices with input–output homogeneity for gate concatenation, which is a way information is processed inside biological cells. In the present instance, the memory component is serially integrated with an INHIBIT gate whose final output is fluorescence emission. It is also remarkable that all the output states of the device are optically self-indicating. The current work also demonstrates a useful supramolecular mechanism for reversible guest capture/release in water by erecting/collapsing phenylene ring walls in a p-cyclophane by chemical redox conversions. Reduction is the set(S) input which gives the output of guest capture. Oxidation is the reset(R) input which produces the output of guest release. The ideas underlying the RS flip-flop are explained in the Discussion section below.

## Results

**Synthesis**. The four compounds required for the systems 1/2 and 3/4 (Fig. 1c) are synthesized as follows. Macrocyclization assisted by $Cs_2CO_3$ produces 1 from 4,4′-dihydroxybenzophenone (5) and 1,5-diiodopentane[37]. $NaBH_4$ reduction[52] and work-up of 1 gives 2. Importantly, 1/2 can be derivatized so that biocompatible water-soluble versions 3/4 become available. This begins with selective monoiodination at ortho-positions with respect to the oxygens of all four phenylene rings of 1 to give 6 in good yield[53]. Each iodo substituent is converted to a carboxylic acid moiety to give 3 in good yield by $Pd^0$-catalyzed carbonylation[54]. $NaBH_4$ reduction in water to the dialcohol tetracarboxylic acid 4 completes the synthesis (Supplementary Methods). The pair of tetracarboxylic acid derivatives 3/4 is used for aqueous solution-phase experiments at pH 10.0. The oxidation reaction (air, hot DMSO)[55] converts dialcohols 2/4 to diketones 1/3. 4 is also oxidized to 3 with $KMnO_4$[52] in water at 40 °C (Supplementary Methods). 1/3 and 2/4 are stable under ambient conditions (Supplementary Methods).

**X-ray crystallography**. 1/2 are switched reversibly between small and large states of their macrocycle cavity by simple chemical redox reactions. $E_{red}$ for 1 is −2.3 V and $E_{ox}$ for 2 is +1.7 V (vs sce, MeCN, 60 °C and DMF, respectively, glassy carbon working electrode, $Bu_4NBF_4$). The large cavity captures aromatic guests, but the small cavity does not, as demonstrated by X-ray crystallography (Fig. 1d–j). Diketone 1 has the small cavity due to flattening of the phenylene rings into the macrocycle plane by π-conjugation with the carbonyl groups. The phenylene rings in dialcohol 2 have no such restrictions and stand orthogonal to the macrocycle plane to create the large cavity to accommodate the benzene guest with edge-to-face interactions.

It is clear from the X-ray crystal structures in Fig. 1d, e, g, h that dialcohol 2 has four erect alkoxyphenylene wall sections as found in classical p-cyclophanes[37], where the wall sections are essentially orthogonal to the mean macrocycle plane. One molecule of benzene is symmetrically held within the macrocycle cavity, with the benzene lying in the mean macrocycle plane. The benzene has edge-face contacts[56] with the four erect alkoxyphenylene wall sections, which drive the ordered benzene inclusion within 2. The 1:1 host:guest stoichiometry means that, when the host is switched to the releasing state, molecular-scale dosing occurs—a crucial feature for chemical signalling with highly active agents[50].

In contrast to those concerning 2, the X-ray crystal structures in Fig. 1f, i, j show that diketone 1 has more-or-less collapsed alkoxyphenylene wall sections that are π-conjugated to the carbonyl groups to some degree. Of each pair, one alkoxyphenylene is aligned much more towards the mean macrocycle plane than the other as a result of the cross-conjugation with the carbonyl. Their orientation with respect to the mean macrocycle plane are 75.85° and 17.67°. The relative orientation of the alkoxyphenylenes and the carbonyl agrees with that of 4,4′-dimethoxybenzophenone[57,58] (Supplementary Methods) showing that these angles are the result of cross-conjugation rather than sterics originating from the macrocycle. It is clear that the macrocycle cavity has undergone substantial contraction in 1 (c.f. 2) owing to the wall collapse, and no benzene occupant is found. Smaller molecules, e.g. dimethylformamide, are known to be included in some related macrocycles[59,60], though in a disordered fashion[59]. Examination of the packing diagrams shows that benzene is found only in the internal cavity of 2 and it is completely absent from 1.

**Spectroscopy in solution**. Since the X-ray crystallography studies clearly showed that the walled 2 contained one benzene occupant whereas the wall-collapsed 1 lost its occupant, we next examine aqueous solution-phase data of solubilized counterparts 4 and 3, respectively. No suitable crystals of 3 and 4 could be grown from solvents in which they were soluble. 4 and 3 are employed at levels which are below their critical aggregation concentrations (Table 1, footnote a). 7–9 (in the form of their dibromide salts) are evaluated as potential occupants. If the geometric fit is suitable, the dications of these p-xylyldiammonium salts (9 being a bicyclo[2,2,2]octane derivative) should be electrostatically attracted to the tetraanionic macrocycle cavity, besides some favourable hydrophobic attractions between the central parts of both the hosts and guests. The p-xylyldiammonium unit is stable under our redox conditions. The $^1H$ NMR spectra of 7 with/without 3 or 4 are shown in Fig. 2. Complexation-induced chemical shift differences (Δδ) are observed for 7 in the presence of dialcohol 4, but not in the presence of diketone 3. The corresponding Δδ values for dialcohol 4 in the presence of 7 are shown in the inset in Fig. 2. The pattern of these Δδ values, which is evidence for an axial inclusion complex in the solution state, are similar to those seen for classical examples of cyclophane inclusion complexes[37,61,62], but with some differences. No significant

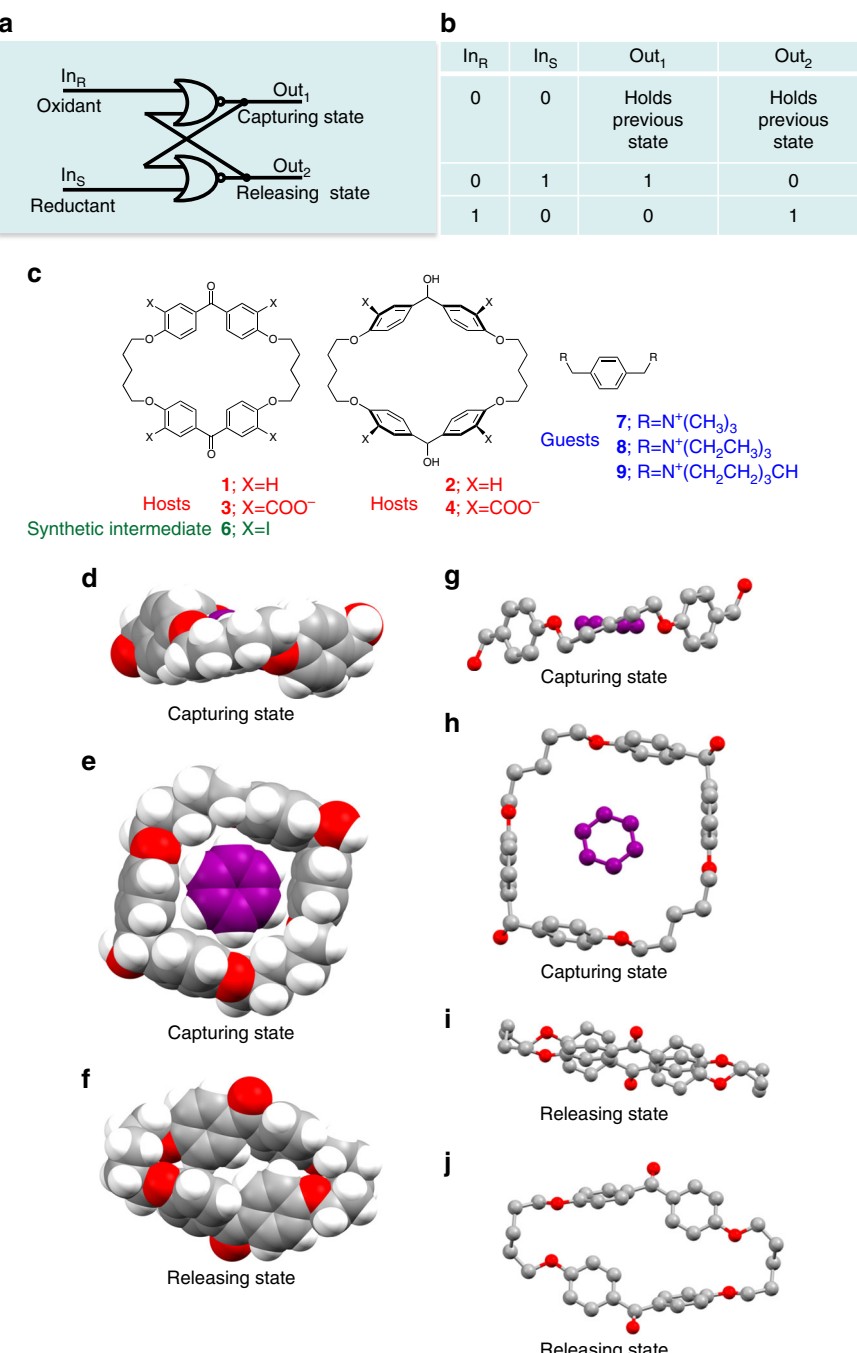

**Fig. 1** Memory, chemical compounds and X-ray crystallography. **a** Physical electronic representation of a computer memory which is conceptually related to the redox-induced reversible guest capture/release by **1/2** in the solid state. **b** Truth table for redox-induced reversible guest capture/release by **1/2**. **c** Molecular structures **1-4** & **6-9** (no stereochemistry is implied). The structure numbers and group identities of hosts, guests and synthetic intermediate are shown in the arbitrary colours of red, blue and green, respectively. **d-j** X-ray crystal structures of **2** (**d**, **e**, **g**, **h**) and **1** (**f**, **i**, **j**) after separate crystallization from benzene. Ball-and-stick representation in elevation view (**g**, **i**), space-fill representation (with calculated hydrogen positions included) in plan view (**e**, **f**) and ball-and-stick representation in plan view (**h**, **j**), are shown in each case, respectively. Case **d** is a space-fill representation (with calculated hydrogen positions included) of **2** shown in elevation view to emphasize the 'wall' aspect of the host

chemical shift differences are found for diketone **3** in the presence of **7**. Somewhat analogous to the X-ray crystallography studies (which employs the smaller and neutral guest benzene), inclusive complexation is clear between **4** and **7**, and no binding is observed for **3** and **7** in spite of ion-pairing in both cases. Although *p*-cyclophane hosts are known to be shape-dependent binders[37], reversible all-or-none switching between binding and releasing states in water has not been demonstrated until now.

The complexation between **4** and **7-9** is quantified by measurement of concentration-dependent $\Delta\delta$ values to obtain binding constants ($\log\beta$) which have values around 4 (Table 1).

Fluorescence spectra also illuminate the interaction between **4** and **7-9**. **7-9** quench the fluorescence of **4** (Supplementary Methods)[63]. Thus, **4** optically self-indicates its guest occupancy status. **4** is excited without significant absorption by **7-9** except at rather high concentrations of the latter. The addition of

self-indication to host **4**'s redox switchability immediately extends its logic integration[64] (Fig. 3a). Since a high level of the guests **7–9** quenches the fluorescence, **7–9** serve as the inhibiting input ($In_4$) to the INHIBIT logic gate (Fig. 3a), which is now integrated to the $Out_1$ line of the RS Flip-Flop in the form of $In_3$ (Fig. 3a, b). This is the bridge between the sequential logic component and the combinatorial component. Such fluorescence quenching is likely caused by photoinduced electron transfer (PET)[65,66] from the alkoxyphenylene units of **4** to the electron-poor phenylene group of **7–9** within the pseudo-intramolecular system. PET from **4** to **7** is thermodynamically allowed, with $\Delta G_{PET} = -0.2$ eV (Supplementary Methods). No exciplex emissions are seen[67]. Analysis of the variation of fluorescence intensity with the concentration of **7–9** gives $\log\beta$ results essentially in agreement with the

corresponding values obtained via [1]H NMR spectra (Table 1). The releasing state has a high $\varepsilon$ of 31,000 $M^{-1}$ $cm^{-1}$ at 304 nm for the $\pi\pi^\star$ band owing to the well-delocalized aromatic ketone units and has a low $\Phi_{Flu}$ of 0.003[68] ($\lambda_{Flu} = 458$ nm) due to energetically low-lying $n\pi^\star$ excited states[69]. The capturing states have low $\varepsilon$ values of 3900 $M^{-1}$ $cm^{-1}$ at 290 nm owing to the smaller $\pi$-delocalization of the benzhydrol moieties, while the $\Phi_{Flu}$ value is low (0.004) when the guest **7** is bound and high (0.013, $\lambda_{Flu} = 356, 416$ nm) when guest-free. All three output states, i.e. vacant capturing state, occupied capturing state and releasing state, are therefore distinguishable by optical spectroscopy.

## Discussion

Having been inspired by the roles played by cyclophanes in supramolecular chemistry[37,42,45,48], we felt that we could apply our experience of molecular switching[4,10–12] to make a contribution to this area. Excellent examples of switchable cyclophanes and related macrocycles already exist[37–44], but binary all-or-none switching of cyclophanes in water still remains a worthwhile goal. Since cyclophanes are well-known to bind shape-complementary organic guests in water[37], a method of switching the cavity size/shape should enable the guest's release. Such guest capture/release would have immediate consequences for dosing of drugs and other bio-signalling agents, as well as environmental beneficiation by toxin removal. Notably, each of these interventions could remain under the user's external control or could be triggered by an existing (e.g. intracellular) condition. The capabilities of such a switchable system could be made more intelligent if it could be endowed with self-indicating properties. Self-indication is key for nanometric molecules to communicate their status to their far-larger human handlers, i.e. to say when a task has been accomplished successfully[18].

Such relatively complex molecular-logic behaviour can benefit from the insights of computer science[4]. Computer scientists and electrical engineers have dealt with such situations in larger-sized systems. This relationship between the disciplines is clear to us since we had the pleasure to introduce the field of molecular-logic-based computation[10].

---

**Table 1** [1]H NMR and fluorescence emission spectroscopic data for complexation of **4** and **7–9**[a]

| Datum | 7 | | 8 | | 9 | |
|---|---|---|---|---|---|---|
| $\delta_{ArH}, -\Delta\delta$ | 7.72 | 0.74 | 7.66 | 0.58 | 7.62 | 0.52 |
| $\delta_{BzH}, -\Delta\delta$ | 4.58 | 0.36 | 4.50 | 0.26 | 4.40 | 0.31 |
| $\delta_{\alpha}, -\Delta\delta$ | 3.14 | 0.29 | 3.27 | 0.22 | 3.46 | 0.42 |
| $\delta_{\beta}, -\Delta\delta$ | — | — | 1.42 | 0.11 | 2.19 | 0.16 |
| $\delta_{\gamma}, -\Delta\delta$ | — | — | — | — | 1.97 | 0.26 |
| $\log\beta^{b}$ | 4.2 | | 3.7 | | 4.6 | |
| $\log\beta^{c}$ | 4.5 | | 4.5 | | 4.6 | |
| $QF^{d}$ | 3.1 | | 2.4 | | 1.7 | |

[a]Compound numbers **4** and **7–9** are given in bold formatting. $D_2O$, pD 10.0 for all experiments except fluorescence spectroscopy. $\delta$ values for **7–9** are given. $\Delta\delta$ values of $-0.02 + 0.02$ for all protons are seen when **3** is used instead of **4**. The uncertainty in this case is the sample-to-sample variability. For details, see Supplementary Tables 1–3. The critical aggregation concentrations (CAC) of **4** and **3** are $1.6 \times 10^{-3}$ and $2.0 \times 10^{-3}$ M, respectively
[b]Determined by [1]H NMR spectroscopy from analysis of $\Delta\delta$ values for $\delta_{ArH}$, according to the equation $(\Delta\delta / \Delta\delta_{max})/[1 - (\Delta\delta / \Delta\delta_{max})]^2 = \beta a$[73], where $a$ is the concentration of **7–9**. 1:1 molar ratios of **4**:**7–9** are maintained in the concentration range $10^{-5}$–$10^{-3}$ M
[c]$H_2O$, pH 10.0. Determined by fluorescence emission spectroscopy from analysis of integrated fluorescence intensity ($I_F$) (excited at 290 nm), according to the equation $\log[(I_{Fmax} - I_F) / (I_F - I_{Fmin})] = \log\beta - pa$[73,74]. $2.5 \times 10^{-5}$ M **4**, $0$–$10^{-3}$ M of **7–9**
[d]Quenching factor (QF) for integrated fluorescence intensity ($I_{Fmax} / I_{Fmin}$)

---

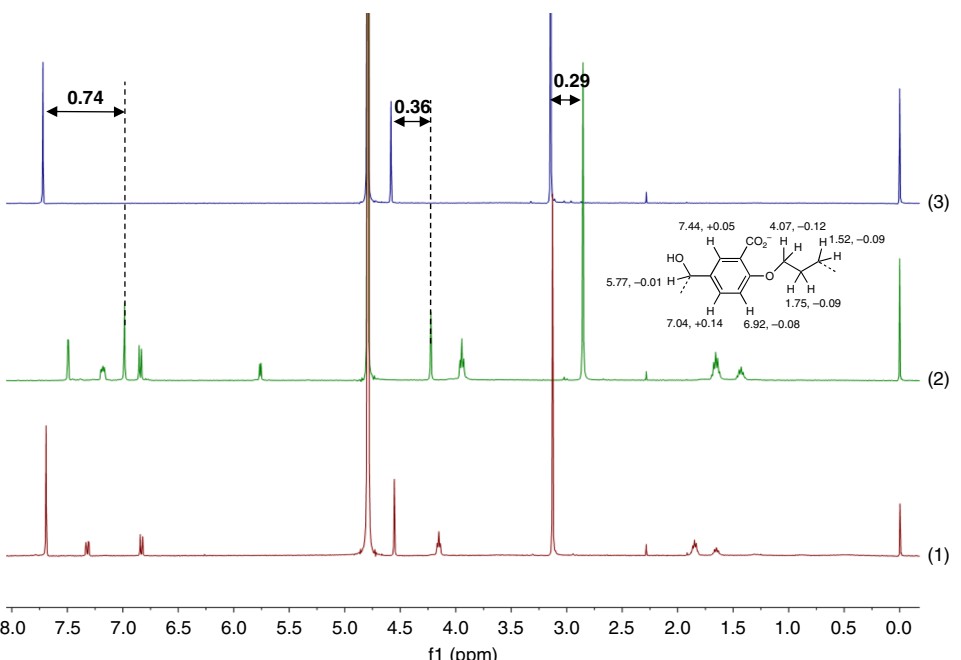

**Fig. 2** [1]H NMR spectral evidence for guest binding/non-binding. [1]H NMR spectra in $D_2O$, pD 10.0. (1) **7 + 3**, (2) **7 + 4** and (3) **7** (all $10^{-3}$ M). $\Delta\delta$ values for **7** caused by **4** are shown. Inset. $\delta$ values in the [1]H NMR spectra of **4** (a part structure is shown) and $\Delta\delta$ values caused by **7**

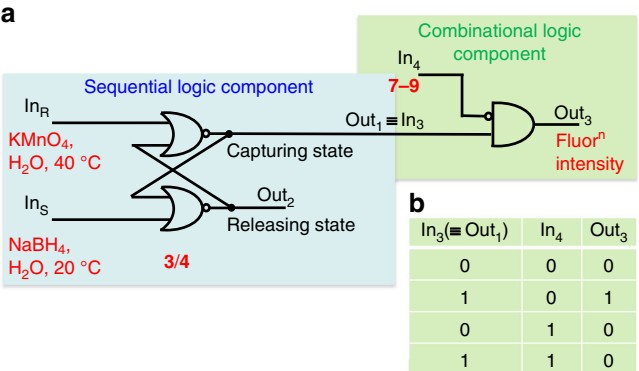

**Fig. 3** Integrated sequential-combinational logic. **a** Physical electronic representation of redox-induced reversible guest release/capture by **3/4** in water followed in the latter case by fluorescence signalling according to INHIBIT logic. Output$_1$ is the status of the capturing state, which is characterized by the UV spectroscopic parameter $\varepsilon = 3{,}900\ M^{-1}\ cm^{-1}$ at the absorption maximum wavelength ($\lambda_{max}$) 290 nm. Output$_2$ is the status of the releasing state, which is characterized by $\varepsilon = 31{,}000\ M^{-1}\ cm^{-1}$ at $\lambda_{max}$ 304 nm. **b** Truth table for guest (**7**)-induced quenching of fluorescence of device **3/4** in water corresponding to an INHIBIT logic gate. Guest-free fluorescence quantum yields ($\Phi_{Flu}$) are 0.013 and 0.003 for **4** and **3**, respectively, at pH 10.0, determined by comparison with 2-methoxybenzoate in water at pH 8.0 ($\Phi_{Flu} = 0.011$) as secondary standard[68]. **7** quenches the fluorescence of **4** by a factor of 3.1. **7** does not measurably affect the fluorescence of **3**. The output is digitized by applying a threshold value of 0.008 to the $\Phi_{Flu}$ values

The inclusive complexation of a guest by a cyclophane can be seen, from some viewpoints, to be a case of a guest being corralled by a surrounding wall, with the corral itself being specified by the macrocycle and with the wall height being determined by the width of the aromatic rings. Notably, the walls are perpendicular to the corral area, i.e. the aromatic ring planes are orthogonal to the mean macrocycle plane. Human experience[70] suggests that a guest cannot be corralled if the walls have fallen down for some reason. The present work shows that these human situations are mirrored in the molecular world.

Although cyclophanes of this type are well-known[37], a prominent feature of our particular cyclophanes is the presence of a pair of ketones or secondary alcohols with flanking phenylene moieties. The benzhydrol version behaves as a more-or-less normal cyclophane host by orienting its phenylene ring planes orthogonal to the macrocycle plane, thereby creating a rather deep and hydrophobic cavity which can include an aromatic guest[37,61]. On the other hand, the benzophenone version has additional cross-conjugation due to the ketones flanked by the phenylene units. This flattens one phenylene of the pair beside each ketone so that the ketone and the phenylene form a plane which is nearly in the macrocycle plane. Such phenylene ring rotation closes off a large part of the macrocycle cavity. Thus a suitable aromatic guest which is inclusively complexed by the benzhydrol version is rejected by the corresponding benzophenone.

It is well-appreciated that ketones and secondary alcohols can be interconverted by appropriate reduction and oxidation[52], each of which are efficient unidirectional chemical reactions. When the ketone is treated with a reducing agent, the corresponding alcohol is formed and it stays as an alcohol whether excess reductant is present or whether the reductant is removed entirely. Similarly, when the alcohol is treated with an oxidant, the corresponding ketone is formed which persists whether the oxidant is absent or present in excess.

From a computer science perspective, the ketone/alcohol system can exist in either of two stable states, depending on the input condition. If a high concentration of reductant is applied as the input, the resulting output state is the alcohol because of the reduction of any starting ketone. This input is called the set(S) input because it sets the alcohol as the output state (Out$_1$ = 1, Out$_2$ = 0 in Fig. 1a, b). If a high concentration of oxidant is applied as the input, the resulting output state is the ketone (Out$_1$ = 0, Out$_2$ = 1 in Fig. 1a, b). This input is called the reset(R) input because it returns the output state to the ketone with which we started. If no input is applied, the system maintains its previous state.

The above behaviour pattern is well-known to computer scientists as a RS flip-flop, which is the memory element in modern computers[4]. Indeed, we have here an all-chemical RS flip-flop, where the inputs, outputs and the device itself are all chemical species. Molecular RS flip-flops based on photochromic[27], enzyme[29], and DNA[28] systems are known. We have pointed out that chemical oxidation and reduction of molecules can also lead to RS flip-flop action[4]. The truth table of a RS flip-flop annotated for our situation is shown in Fig. 1b. However, it is important to note some key differences between the electronic and chemical RS flip-flops arising from the diversity present within chemistry. The electronic version is prevented from receiving two high inputs (hence the truth table having only three rows) since it leads to an unstable output situation. The chemical version naturally avoids receiving two high inputs because it is recognized that mixing equivalent amounts of reductants and oxidants would not have a net effect on a substrate. Secondly, the two outputs of an electronic RS flip-flop are always opposite whereas the chemical case is more nuanced. Opposition is available in terms of the guest-capturing and -releasing states. Opposition is also available in terms of the ketones and the alcohols being redox-interconvertible. On the other hand, there is no natural opposition between ketones and alcohols per se or in terms of their different chemistries.

In conclusion, X-ray crystallography shows that, when $\pi$-conjugation opportunities are depleted in benzhydrol derivative **2**, the phenylene units in a *p*-cyclophane orient essentially orthogonal to the mean macrocycle plane[37]. This macrocyclic cavity encapsulates benzene. When extra $\pi$-conjugation is arranged via ketone groups in benzophenone derivative **1**, the phenylene groups flatten significantly into the mean macrocycle plane so that no encapsulation is possible. The aqueous solution-phase spectroscopy experiments on **4** and **3** confirm that inclusive macrocycle occupancy (by **7–9**) is only permitted for the benzhydrol **4** with its erected phenylene walls. An all-chemical RS flip-flop feeding an INHIBIT gate is demonstrated since chemical redox inputs applied to the *p*-cyclophane devices **1/2** or **3/4** permit reversible all-or-none capture or release of the guests where guest occupancy is self-indicated by the fluorescence output being quenched. This augurs well for the small-scale serial integration of sequential and combinational logic devices in the context of guest capture/release applications.

## Methods

**Synthesis and characterization of the molecular-logic gate arrays and their precursors.** All the procedures are given in the Supplementary Methods.

**X-ray crystallography of the guest-binding and guest-releasing states.** Single crystals of **1** and **2** were obtained by evaporation of concentrated solutions of the compounds in benzene. Suitable crystals were selected and measured on a Rigaku Saturn724 + (2 × 2 bin mode) diffractometer. The crystals were kept at 100 K during data collection. Using Olex2[71], the structures were solved with the ShelXS[72] structure solution programme using direct methods and refined with the ShelXL[72] refinement package using least squares minimization. X-ray crystallographic data

for **1** and **2**, as well those for control compounds are detailed in the Supplementary Methods.

**Guest-dependent fluorescence spectroscopy of the dialcohol cyclophane.**
Fluorescence spectroscopic data obtained on a Perkin-Elmer LS55 spectrometer running FLWinlab software for **4** as a function of added guest are detailed in the Supplementary Methods.

## Data availability

The authors declare that the data supporting the findings of this study are available within the Article and its Supplementary Information files. The X-ray crystallographic coordinates for structures **1** and **2** reported in this study have been deposited at the Cambridge Crystallographic Data Centre (CCDC), under deposition numbers 1875556–1875557. These data can be obtained free of charge from The Cambridge Crystallographic Data Centre via www.ccdc.cam.ac.uk/data_request/cif.

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

## Acknowledgements

We thank the Department of Education and Learning of Northern Ireland, Engineering and Physical Sciences Research Council of UK, China Scholarship Council, Queen's University Belfast, Dr. A.P. Doherty, Dr. W.D.J.A. Norbert, C. Kelly, Z.Q. Chen and L.F. Gui for support and help. In Memoriam Professor John McGarvey, Professor Ken Seddon and Dr. Sergiy Shekhtman.

## Author contributions

H.Q.N.G. designed the synthesis. B.D., T.S.M., A.J.M.H., C.Y.Y., B.S. and H.Q.N.G. contributed to the synthesis, optimization and characterization. B.D., A.A.-A., J.F.M. and P.N. carried out the X-ray crystallography. B.D., T.S.M. and A.J.M.H. conducted binding studies and analyzed data. C.Y.Y. assisted during the revision. A.P.deS. planned the research, analyzed data and wrote the manuscript.

## Additional information

**Competing interests:** The authors declare no competing interests.

