## [Peer Review File · Nature Communications]

Reviewers' comments:

Reviewer #1 (Remarks to the Author):

The communication reports a series of dibenzophenone cyclophanes that can undergo reversible chemical reduction/oxidation. The conformational flexibility of at least one oxidised example provides a cavity that nicely accommodates small aromatic guest molecules, such as benzene in 2 (but not 1), as evidence by single crystal X-ray crystallography. This aspect of the work is interesting to researchers in the field of supramolecular chemistry and guest-host molecular recognition. The authors additionally examine the interaction of p-xylyldiammoniums (p-dibenzyl derivatives) as guests by ¹H NMR (no crystal structures could be grown) and fluorescence spectroscopy and observe an emission 'turn-off' response only with benzhydryl macrocycle 4, notably in water. The study of such supramolecular interactions is of current interest, as for example, a cyclophane with perylenebisimides in chloroform: A Perylene Bisimide Cyclophane as a "Turn-On" and "Turn-Off" Fluorescence Probe *Angew. Chem. Int. Ed.* 2015, 54, 10165–10168. <https://doi.org/10.1002/anie.201503542>

Furthermore, the interpretation of the chemistry from a logic perspective involving the integration of a sequential logic array (flip-flop) and a combinatorial logic array (INHIBIT logic gate) illustrates a conceptually different way of dealing with the issue of input-output homogeneity and gate concatenation. Additionally, this is done in water for 3/4 which foreshadows practical uses in biological and environmental applications. I believe that researchers in molecular logic-based computation will be influenced by the insights provided by this paper.

As I was reading through the manuscript, I recalled a paper by de Silva written almost 20 years ago, which introduced the concepts of functional integration and sequential operation at the molecular level and the first three-input INHIBIT logic gate (*J. Am. Chem. Soc.* 1999, 121, 1393-1394). I feel this paper should be included in the manuscript. It leads me to inquire: Given that the cyclophane incorporates benzophenone moieties (particularly the intermediate with iodine), did the authors perform phosphorescence measurements and consider the integrated sequential/combinatorial system resulting at this longer time domain?

Some additional queries:

1. In Fig. 1 should the input for Ln 4 be 7-9 rather than just 7? An apparent weakness in Fig. 1 is that the flip-flop for 3/4 is in water and the fluorescence studies with 7-9 are in water. However, 1/2 are oxidised in DMSO and reduced in THF. It would perhaps make a stronger case to highlight only the results in aqueous solution in Fig. 1.
2. Fig. 2: The resolution of the left side of the figure 2 notably labels 1-8 should be improved.
3. What is the fluorescence response at lower physiological pH of 7.4? Can the authors provide the fluorescence titration curves to supplement footnote c in Table 1?
4. In Table 1: define QF footnote d.
5. Methods, synthesis: The number of significant figures for the mass and moles are not consistent.
6. Figures S1-S3: Add an arrow pointing downward beside fluorescence curves and the compound number (i.e. arrow [7]).
7. Line 55: Should it read: NaBH₄ reduction and work-up of 1 and gives 2.
8. Line 67: Are the standard potentials experimental values (then give electrode material) or literature values (give references).
9. Do the authors have UV-visible absorption spectra to include in the SI?
10. What standard was used to measure the relative quantum yields reported in Figure 1 caption. Given that the QYs are understandably low in water, were any studies also performed in mixed aqueous/alcohol media?

Reviewer #2 (Remarks to the Author):

The contribution of de Silva and co-workers describes a new concept in the development of molecular logic devices. The work is very interesting and clearly will be a major addition to the field; which is fast growing, as recently demonstrated with a review article in Chemical Society Reviews (NB, missing from this contribution). While the logic operations are centred around molecular motion that enables the opening and closing of a cyclophane, and the inputs are both chemical and light (allowing down-processing which is an important development), then my main concern is that the manuscript has been written in very concentrated manner and lacks clarity. The issues with this is that the technical specialised language is not assessable to the general readership of Nature Communications, and it feels like that there is too much of assumed knowledge, and this has to be rectified. As the 'communication' is quite short, I suggest that the authors incorporate some of the figures from the Supporting information (particularly the changes in the ground state, etc.) and elaborate more on the mechanism observed, the truth tables, to elevate the impact of this work, and explain the diagram in Figure 1 better. The science here is sound, it is more that the authors need to communicate their finding in a more accessible way; as an example, then 'the wall being erected' or 'collapsed' needs to be more clearly connected with the structural changes occurring. Hence, the view of the X-rays need to be side-ways so this is clearly visible to the reader and more connected with the flow of thoughts of the authors. Another example is the use of the 'set(S)' and 'rest(R)' terminology, which needs to be better explained and the starting states etc. needs to be better described, as the logic operation in Figure 1a is not trivial. I suggest that the synthesis is also moved into the supporting information as having it as part of the main text, takes away from the flow of the main emphasis (e.g. how the cyclophanes were made is not of interest here, more so how the opening and closing of their cavity can results in capture and relapse of the host, and hence, in a 'concatenated' process), etc. I also think that Figure 1 needs to be introduced later in the article, or re-distributed within the article, and then summarised at the end, as the outcome of these results are shown before we are introduced to the compounds in Figure 2, and the general discussion. Overall this is a very novel, difficult and important concept that is introduced by the de Silva group; however, it needs to be more clearly presented, and that is easily achievable, after which, I am happy to recommend the acceptance of this work for publication in Nature Communications.

Reviewer #3 (Remarks to the Author):

Crystallography Report

I suggest one rather cosmetic change to the cif of the keto form: There is an e.s.d. given to the 90-degree-angles which in my opinion should be deleted. Otherwise I see nothing wrong with the crystal structures.

Reviewers' comments:

Reviewer #1 (Remarks to the Author):

The communication reports a series of dibenzophenone cyclophanes that can undergo reversible chemical reduction/oxidation. The conformational flexibility of at least one oxidised example provides a cavity that nicely accommodates small aromatic guest molecules, such as benzene in 2 (but not 1), as evidence by single crystal X-ray crystallography. This aspect of the work is interesting to researchers in the field of supramolecular chemistry and guest-host molecular recognition. The authors additionally examine the interaction of p-xylyldiammoniums (p-dibenzyl derivatives) as guests by H NMR (no crystal structures could be grown) and fluorescence spectroscopy and observe an emission 'turn-off' response only with benzhydrol macrocycle 4, notably in water. The study of such supramolecular interactions is of current interest, as for example, a cyclophane with perylenebisimides in chloroform: A Perylene Bisimide Cyclophane as a 'Turn' \On and 'Turn' \Off' Fluorescence Probe *Angew. Chem. Int. Ed.* 2015, 54, 10165-10168. <https://doi.org/10.1002/anie.201503542>

Response: This reference has been added to the revised manuscript as ref. 64.

Furthermore, the interpretation of the chemistry from a logic perspective involving the integration of a sequential logic array (flip-flop) and a combinatorial logic array (INHIBIT logic gate) illustrates a conceptually different way of dealing with the issue of input-output homogeneity and gate concatenation. Additionally, this is done in water for 3/4 which foreshadows practical uses in biological and environmental applications. I believe that researchers in molecular logic-based computation will be influenced by the insights provided by this paper.

Response: We are delighted that Reviewer #1 appreciated what we tried to say.

As I was reading through the manuscript, I recalled a paper by de Silva written almost 20 years ago, which introduced the concepts of functional integration and sequential operation at the molecular level and the first three-input INHIBT logic gate (*J. Am. Chem. Soc.* 1999, 121, 1393-1394). I feel this paper should be included in the manuscript.

Response: We thank Reviewer #1 for noticing this old reference. We happily include it as ref. 65. However, we need to point out that our meaning of 'sequential operation (of logic gates)' in that reference was the general English meaning, i.e. that different gates can be used at different times, e.g. when different lines of a program calls out different subroutines to engage different gates. There were no memory elements in that reference. On the other hand, the accepted meaning of 'sequential logic operation' among computer scientists and molecular computing practitioners is that the gate itself is time-dependent or history-dependent or capable of memory.

It leads me to inquire: Given that the cyclophane incorporates benzophenone moieties (particularly the intermediate with iodine), did the authors perform phosphorescence measurements and consider the integrated sequential/combinatorial system resulting at this longer time domain?

Response: We thank Reviewer #1 for noticing this phosphorescence possibility for the benzophenone moieties, and particularly those containing heavy atom iodine. Such a possibility was explored in ref. 65 by including the phosphor unit containing heavy atom bromine within a protective cyclodextrin to prevent triplet-triplet annihilation in water. Unfortunately, we have no such protection available for our benzophenone-based cyclophanes. Therefore no phosphorescence data could be obtained in the present cases.

Some additional queries:

1. In Fig. 1 should the input for Ln 4 be 7-9 rather than just 7?

Response: We agree with Reviewer #1's point. The correction has been made in Fig. 2.

An apparent weakness in Fig. 1 is that the flip-flop for 3/4 is in water and the fluorescence studies with 7-9 are in water. However, 1/2 are oxidised in DMSO and reduced in THF. It would perhaps make a stronger case to highlight only the results in aqueous solution in Fig. 1.

Response: We agree with Reviewer #1's point. The revision accommodates this and Reviewer #2's wishes by showing the RS flip-flop alone for the solid-state case where the redox reactions are non-aqueous in Fig. 1. Later, we show the concatenated gate system for only the results in aqueous solution in Fig. 2.

2. Fig. 2: The resolution of the left side of the figure 2 notably labels 1-8 should be improved.

Response: This loss of resolution was a doc-pdf file conversion issue and is not a problem in the original doc file. Nevertheless, this apparent problem has been mitigated by using a larger font-size in the original picture.

3. What is the fluorescence response at lower physiological pH of 7.4?

Response: Unfortunately, the compounds are not sufficiently soluble in a non-aggregated form at this pH value to perform meaningful experiments. We hope to overcome this problem in the future by synthesizing the corresponding cyclophane sulfonic acids.

Can the authors provide the fluorescence titration curves to supplement footnote c in Table 1?

Response: We have added these within section S5.

4. In Table 1: define QF footnote d.

Response: Quenching factor (QF) has been clarified in footnote d.

5. Methods, synthesis: The number of significant figures for the mass and moles are not consistent.

Response: These have been adjusted so that mass and moles have similar uncertainties. Most molar quantities are given as mmol values for added consistency. Please note that the methods and synthesis procedures have been shifted to the Supporting Information to satisfy a suggestion by Reviewer #2.

6. Figures S1-S3: Add an arrow pointing downward beside fluorescence curves and the compound number (i.e. arrow [7]).

Response: Done in all three cases.

7. Line 55: Should it read: NaBH₄ reduction and work-up of 1 and gives 2.

Response: Done.

8. Line 67: Are the standard potentials experimental values (then give electrode material) or literature values (give references).

Response: These are experimental values obtained with a glassy carbon electrode. This has been added in the revision.

9. Do the authors have UV-visible absorption spectra to include in the SI?

Response: This is done as section S4. Unfortunately, the guest-induced UV-visible absorption spectral changes are not significant enough for meaningful analysis even at low enough guest concentrations. We had hoped for the UV-visible absorption spectral changes to mirror the fluorescence spectral changes, but it appears that (a) the π -electron systems of the host and the guest are not parallel enough to permit a π - π interaction in their ground states and (b) the electron richness of the host and the electron deficiency of the guest are not large enough to cause an element of charge transfer. Point (a) is supported by the $\Delta\delta$ values induced in the host by the guest (Figure 2b).

10. What standard was used to measure the relative quantum yields reported in Figure 1 caption.

Response: 2-Methoxybenzoate (quantum yield = 0.011, ref. 69) was employed as a secondary standard. This information is added on page 15 as footnote d of Fig.2.

Given that the QYs are understandably low in water, were any studies also performed in mixed aqueous/alcohol media?

Response: While we agree with Reviewer #1 that our understanding of these systems would be improved if fluorescence studies were also performed in mixed aqueous/alcohol media, we restricted ourselves to neat aqueous media since our emphasis was on the switching of these systems in neat water. Hopefully, a fuller understanding of these and related systems will include mixed aqueous/alcohol media during future studies.

Overall, we appreciate Reviewer #1's careful reading and insightful criticism of our paper which has helped us to think more about our results and to produce a revised form which has extra raw data and clearer presentation.

Reviewer #2 (Remarks to the Author):

The contribution of de Silva and co-workers describes a new concept in the development of molecular logic devices. The work is very interesting and clearly will be a major addition to the field; which is fast growing, as recently demonstrated with a review article in Chemical Society Reviews (NB, missing from this contribution).

Response: This reference has been added as ref. 9. We are delighted that Reviewer #2 appreciated what we tried to say.

While the logic operations are centred around molecular motion that enables the opening and closing of a cyclophene, and the inputs are both chemical and light (allowing down-processing which is an important development), then my main concern is that the manuscript has been written in very concentrated manner and lacks clarity. The issues with this is that the technical specialised language is not assessable to the general readership of Nature Communications, and it feels like that there is

too much of assumed knowledge, and this has to be rectified. As the 'communication' is quite short, I suggest that the authors incorporate some of the figures from the Supporting information (particularly the changes in the ground state, etc.) and elaborate more on the mechanism observed, the truth tables, to elevate the impact of this work, and explain the diagram in Figure 1 better. The science here is sound, it is more that the authors need to communicate their finding in a more accessible way; as an example, then 'the wall being erected' or 'collapsed' needs to be more clearly connected with the structural changes occurring. Hence, the view of the X-rays need to be side-ways so this is clearly visible to the reader and more connected with the flow of thoughts of the authors. Another example is the use of the 'set(S)' and 'rest(R)' terminology, which needs to be better explained and the starting states etc. needs to be better described, as the logic operation in Figure 1a is not trivial. I suggest that the synthesis is also moved into the supporting information as having it as part of the main text, takes away from the flow of the main emphasis (e.g. how the cyclophanes were made is not of interest here, more so how the opening and closing of their cavity can results in capture and relapse of the host, and hence, in a 'concatenated' process), etc. I also think that Figure 1 needs to be introduced later in the article, or re-distributed within the article, and then summarised at the end, as the outcome of these results are shown before we are introduced to the compounds in Figure 2, and the general discussion. Overall this is a very novel, difficult and important concept that is introduced by the de Silva group; however, it needs to be more clearly presented, and that is easily achievable, after which, I am happy to recommend the acceptance of this work for publication in Nature Communications.

Response: We have happily made all the adjustments that Reviewer #2 has pointed out. A substantial section has been inserted after the abstract and the leader paragraph (these two have been left unchanged since they are summaries of the work) so that the flow of ideas is laid out more logically. This makes the following sections more understandable. The revised Figure 1 contains most of the changes in the ground state. The original Figure 1 has been changed so that only the memory element is shown in the revised Figure 1. The integration of the memory element with the INHIBIT gate is presented later in revised Figure 2 after the fluorescence quenching data have been discussed. The original side-ways view of the X-rays in ball-and-stick representation is now augmented with a side-ways view of the X-rays in space-fill representation in the revised Figure 2 to show the wall aspect more clearly so that the guest is now virtually invisible behind the 'wall'. The set-reset ideas are described in more detail so that the technical language of computer science is explained in the chemical context. The synthesis procedures are now moved into the supporting information as the beginning of section S1. The synthetic procedures given previously in the supporting information for the guests have now been reformatted for consistency. We are grateful to Reviewer #2 for helping to make our paper clearer and more accessible to the general readership of Nature Commun. as a result.

Reviewer #3 (Remarks to the Author):

Crystallography Report

I suggest one rather cosmetic change to the cif of the keto form: There is an e.s.d. given to the 90-degree-angles which in my opinion should be deleted. Otherwise I see nothing wrong with the crystal structures.

Response: We are happy that Reviewer #3 found our dataset to be essentially sound. We would have happily deleted the e.s.d. but unfortunately the co-authors who are currently available seem to be unable to make this adjustment to the cif file. So please allow us to leave the cif file unchanged.

REVIEWERS' COMMENTS:

Reviewer #1 (Remarks to the Author):

The authors have adequately addressed my queries and curiosities. On review of the revised manuscript, I have a few minor suggestions and considerations, which I believe will strengthen the paper further. Otherwise, I congratulate the authors on this timely paper, which from the author's list, suggests it has been a long time coming.

1. Page 4, line 1: To be more inclusive after "computer scientists" add "and electrical engineers".
2. Page 5, Figure 1: Would molecules 3 and 4 be better represented as the carboxylates rather than the carboxylic acids, given that in Table 1 (footnotes), measurements were performed at pH 10?
3. Page 5, Figure 1: It may not be immediately obvious to readers from the formula of molecule 9 that it is a bicyclo[2,2,2]octane derivative. On page 9, a sentence could be added to further describe 7-9.
4. Table 1 footnote: In the Henderson-Hasselbalch equation, it should be "pKa" rather than "pa".
5. Table 1 footnote: Amend footnotes "a" and/or "c". Footnote "a" refers globally to D2O while "c" refers locally to H2O.
6. Page 10, end of paragraph: Can any comparisons be made with respect to "2 and 7-9"? In other words, do the authors have fluorescence data for 1/2 and 7-9?
7. In Figure 1, to assist the reader, can the authors perhaps explicitly indicate which images represent "on" and "off" states for easy referral?

Point-by-point response to issues raised by referees.

REVIEWERS' COMMENTS:

Reviewer #1 (Remarks to the Author):

The authors have adequately addressed my queries and curiosities. On review of the revised manuscript, I have a few minor suggestions and considerations, which I believe will strengthen the paper further. Otherwise, I congratulate the authors on this timely paper, which from the author's list, suggests it has been a long time coming.

Response: We are grateful to Reviewer #1 for giving up yet more time to re-review and for continuing to improve our paper. Many, many thanks.

1. Page 4, line 1: To be more inclusive after "computer scientists" add "and electrical engineers".

Response: Done

2. Page 5, Figure 1: Would molecules 3 and 4 be better represented as the carboxylates rather than the carboxylic acids, given that in Table 1 (footnotes), measurements were performed at pH 10?

Response: Done

3. Page 5, Figure 1: It may not be immediately obvious to readers from the formula of molecule 9 that it is a bicyclo[2,2,2]octane derivative. On page 9, a sentence could be added to further describe 7-9.

Response: Done

4. Table 1 footnote: In the Henderson-Hasselbalch equation, it should be "pKa" rather than "pa".

Response: Many thanks for drawing attention to this but the term is indeed 'pa'. I apologize for my choice of symbol 'a' for guest concentration (which others have used many times before) that led to this apparent error in the eyes of Reviewer #1. 'p' stands for the usual '-log'. The term corresponding to 'pK_a' in this case is 'logβ'. There is no 'pK_a' in this case since we are dealing with a host-guest binding equilibrium rather than a protonation equilibrium.

5. Table 1 footnote: Amend footnotes "a" and/or "c". Footnote "a" refers globally to D2O while "c" refers locally to H2O.

Response: Well-spotted. This has been corrected.

6. Page 10, end of paragraph: Can any comparisons be made with respect to "2 and 7-9"? In other words, do the authors have fluorescence data for 1/2 and 7-9?

Response: We thank Reviewer #1 for this thought, which will be part of follow-up experiments in non-aqueous media in the not-too-distant future. We have not conducted fluorescence experiments for 1/2 and 7-9 so far, partly because our keenness to confine

ourselves to experiments in water. **1** and **2** are not soluble in water even sparingly.

7. In Figure 1, to assist the reader, can the authors perhaps explicitly indicate which images represent “on” and “off” states for easy referral?

Response: This has been done, by labelling each x-ray structure as either ‘capturing state’ or ‘releasing state’. We thank Reviewer #1 for this very useful suggestion.